# The Significance of Digital Network Platforms to Enforce Musicians' Entrepreneurial Role: Assessing Musicians' Satisfaction in Using Mobile Applications

**Ofilia Psomadaki [1,*], Maria Matsiola [2]**, **Charalampos A. Dimoulas [3]** **and George M. Kalliris [3]**

[1] School of Music Arts and Science, University of Macedonia, 156 Egnatia Street, 54636 Thessaloniki, Greece
[2] Department of Communication and Digital Media, University of Western Macedonia, 52100 Kastoria, Greece; mmatsiola@uowm.gr
[3] Faculty of Economic and Political Sciences, School of Journalism and Mass Communications, Aristotle University of Thessaloniki, 54636 Thessaloniki, Greece; babis@eng.auth.gr (C.A.D.); gkal@jour.auth.gr (G.M.K.)
[*] Correspondence: opsomadaki@uom.edu.gr; Tel.: +30-69-7604-8980 or +30-23-1087-0734

**Abstract:** Digital entrepreneurship through the employment of smartphones and other handheld device applications is an innovative form of customer approach. Particularly, in the cultural marketing sector, new technologies, such as social media, YouTube channels, and mobile applications may increase the artists' visibility and attract new partnerships and audiences. In this framework, entrepreneurs-musicians who attended a seminar on Management of Cultural Heritage, Communication and Media in Greece were asked to promote their activities through the creation of a smartphone application. After having completed their endeavor and further evaluated it, they participated in qualitative research based on the theory of experts' interviews, as a case study. The aim of this paper, through thematic analysis of the conducted interviews, is to reflect upon the dynamics of new technologies in music entrepreneurship. As derived by the analysis, the use of mobile applications may effectively approach prospective partnerships and audiences, especially if combined with other contemporary forms of communication, and results in presenting enhanced professionalism. Audience engagement, which is an issue that is sought while attempting to optimize promotion processes, may be achieved if a further practice is performed. As the research was conducted during the COVID-19 pandemic, the demand for this form of making publicly known their artwork was considered essential.

**Keywords:** mobile applications; musicians' entrepreneurship; digital networks; music distribution; artwork distribution; disintermediation

## 1. Introduction

Social networks, new digital artifacts, and digital platforms are used to pursue an attractive career option with innovative and entrepreneurial opportunities. This fact leads to an increase in the interest in digital entrepreneurship [1,2]. Different types of networking are getting used by firms' performance, which is included in the creative industries [3]. A new venture is getting started by entrepreneurs that use and develop digital technologies or the Internet and cultural commodities circulate with the use of the emerged infrastructures. Platforms, which are ubiquitous in our lives as we buy things, get informed or educated through them, can exert significant influence in shaping the cultural content that users discover and share by using a single device, like a smartphone.

Platformization is a term adopted by Helmond (2015) [4] which in the cultural production domain refers to the reorganization of practices in the production and circulation of cultural products through the launch and great penetration of platforms. In this new environment, artists can become popular and gain visibility while their fans can get access

to their products [5]. The innovations that digital platforms offer enable artists to be more creative according to customer demands and fondness. Music distribution has shifted from material records to digital data reconfiguring the role of music which is available via networks and portable devices and nowadays it can be distributed more easily by technological advances [6]. Music production and creation can be more independent as well propelling the economic entrepreneurial dimension. Spotify, YouTube, App Stores, and many other platforms offer a wide range of usability capabilities that have shaped the role of music in the market. Content presentations such as interface organization and navigation, and algorithmic recommendations that help users to find information on their favorite artist or to propose human-curated or editorial suggestions are a few of them [7]. Digital entrepreneurs use social media platforms through smartphones and other handheld devices and expand their professional and social networks with music fans and collaborators. After all, nowadays, technologies are not considered mere devices, they create socio-technical relationships as well. Furthermore, a variety of modalities are used to minimize the geographic distance between the creators and the artists [8,9]. Social networks allow music fans, and probable partners to share content with music enterprises. Not only this, but also users can share opinions, feelings, ideas, and educational experiences. The quality of music performance, production, distribution, and communication has been improved by the addictive cyber relationship of blockchain technology, virtual reality, and artificial intelligence [10–12].

As Pietila (2019) and Waldfoel (2021) mention, musicians–entrepreneurs became truly independent since digital technologies help them to enter the music industry easier in comparison to the past and provide them with a variety of tools, especially mobile applications [13,14]. By employing contemporary applications, Kalliris, Dimoulas, and Matsiola (2019) claim that many processes which are related to music can nowadays be performed in digital home studios [15]. Music entrepreneurs can handle hardware with the use of music applications and that fact led to the emergence of a wider spectrum of talents because of the reduced expenditures. Traditional barriers of cost and skills have been removed, according to Dewenter et al. and Karubian [16,17] by technological developments, as direct exchanges between producers, who can provide their products while generating revenue, and buyers, who are benefited from alternative options and lower prices, are supported [18]. Musicians, nowadays, may earn retribution (financial and recognizability) easier than in any other era. Specifically, their fans, via their handle head devices, such as mobile phones, through registration to applications or platforms such as Apple iTunes, may pay for the musical creations. Moreover, musicians also gain international perceptibility without geographical boundaries [19].

The learning management system is one of the methods and practices that researchers, in a framework of extending traditional heuristics for evaluating specific applications in distinct areas, are proposing [20,21]. As many existing research works aim in this direction, the importance of developing mobile applications as a teaching method and as a useful tool for entrepreneurs is indicated [22]. The authors' academic research interest lies in technology (sound, smartphone applications, etc.), culture (in the domain of cultural heritage as well), and technology-enhanced education [23–29] and comprehends that combined technological developments and entrepreneurial skills are important factors in modern economies. The authors were engaged in this research, under the prism of the contemporary social and business environment where the adoption of smartphone applications, which are becoming simple and natural to use, may lead to successful entrepreneurial endeavors, even when created by amateurs, though under proper educational training and guidance. The motivation of this work was initiated by the noticed urge among young people to find a way to express their creativity in innovative ways as entrepreneurs in a mature field such as mobile applications to create audience awareness and engagement. Through their unique perspectivity regarding problem-solving of the activities of their own work after being challenged to think about their needs in relation to their audience and prospective

customers, and the feedback they may offer, contemporary forms of pedagogical models could transform as be focused to reach a higher level of successful education.

In this framework, the present investigation, as a case study, attempted to answer in a prime stage, whether mobile applications, as tools that could aid in building a career path, hold a significant role in entrepreneurs' promotion through an educational, though addressed to professionals, process.

Therefore, the research questions that this paper seeks to reflect upon -as well as raise- mainly concern the following, expressed as research questions:

RQ1: Are new technologies effective in promoting the work of young entrepreneurs in the music industry?

RQ2: Are smartphone cultural applications helpful for entrepreneurs related to music?

RQ3: Are web communication technologies favored by young entrepreneurs in publicizing their work?

RQ4: During the COVID-19 period, were the communication technologies thought of as helpful in promoting their artwork by the young entrepreneurs in the music field?

In the following sections, a literature review will state the former and up-to-date published research works which are related to the significance of digital network platforms. Especially, those focused on the use of mobile applications as a useful tool for entrepreneurs who want to promote their work in the music industry. Consequently, the methodology followed will be presented in Section 2 (Materials and Methods). Afterward, the results of the qualitative research will be analyzed in Section 3 (Results). Finally, in the discussion section, the authors' findings will be highlighted, and the limitations of the study will be stated along with future work suggestions.

*Literature Review—New Technologies in Music Entrepreneurship*

Entrepreneurship related to culture combines different domains of action and knowledge, encompassing economic and cultural features, that both concern Sciences in the field of Anthropology and Sociology [30]. Cultural entrepreneurs, mostly young people who are active members of society, as Scott (2012), defines them, mainly aim to build a career in the art domain. According to Scott (2012), cultural entrepreneurs are those people who create new cultural products and try to establish their identity at low cost, using innovative methods that technology offers them [22].

Uncertainty and complexity of the economy at a global level are augmenting. This fact leads to the need for an entrepreneurial response [31] that would help to build a behavioral control [32,33] by motivating people to be more active in themes related to culture, especially music [34], by promoting them to build not only positive attitudes towards entrepreneurship but enhanced perceived behavioral control [35,36].

Artists and producers, who are trying to make their own music brand name according to Swedberg (2006) [36] seek guidance and qualification activities to find their entrepreneurial roles. These activities include discussion with professionals and strategies for innovative production, marketing, and distribution. As a result, incubators began to emerge in the music field. The business incubator model is being used in new companies to support their development by providing new services, training, or bigger office space. Similarly, in the music domain, music incubators help the artist's evolution through their entrepreneurial role by meeting new demands. For instance, some of the musicians also work as teachers, producers, and sound engineers in the case of incubators [37].

In the last decade, the innovations that network technology and the Internet brought into our daily lives, such as mobile applications, have completely changed the global economic activities and business landscape [38] developing e-commerce. The employment of contemporary communication technologies provides new approaches to the manufacturer to reach the end-user while "reducing the distance" as they provide new forms of interaction [39]. These changes affected the cultural sector, including the music business which has gone under a significant shift that impacted the forms of production, distribution, and consumption [18] introducing new modes of entrepreneurs who addressed

fragmented audiences. Cultural and Creative Industries (CCI) and technology advances create new opportunities and challenges and form new business models, platforms, and processes, especially for small businesses, which comply with the contemporary modes of cultural product distribution [40,41] and deliver to the audience. Digital technologies are omnipresent and offer solutions to explore new possibilities in a constantly evolving cultural entrepreneurship by people who present a capacity to adapt to unforeseen events [42]. Active social media and digital music broadcasting platforms employment, as well as innovative ways of adoption to gain visibility, are considered as motivation to building a unique style [42]. In addition, cultural social networks that musicians join every day help them to improve their careers through the advice of successful music entrepreneurs. The result of cooperation between musicians and entrepreneurs using these networking services [39], led to the creation of cultural music incubators and alliances that advance the businesses and prospective careers. In the contemporary technologically mediated societies where technical materiality has shifted into communication processes, people interact through infrastructures and new technologies facilitate the creation of collaborations globally as engagement platforms. Actors intensify the exchange of artistic, technological, and co-creation resources [43] and knowledge thus being effective in promoting artwork in extended networks [44]. The emergence of "Music 2.0" allowed musicians and entrepreneurs who are related to music to connect directly to their audiences. Moreover "Music 2.0" allowed them to contact their audience from a position previously occupied by recording brands. Young and Collins (2010) [45] were the first who paraphrased the term Web 2.0 and used the term "Music 2.0", to emphasize the dynamic role that Web 2.0 offered to music. "2.0" is used to describe the second generation of the web which, through its design and features gives the opportunity to anyone to create and share online information they have created. Additionally, it makes communication and collaboration among users easier. According to Young and Collins (2010) "Music 2.0", is a new media environment that offers musicians the opportunity to create and distribute content related to music with unprecedented ease [45].

Although "Music 2.0" provides a variety of easy-to-implement services, however, there are some factors that should be taken into account. For instance, there is a considerable need for time and effort for musicians to build individual relationships and complex negotiations to accomplish the news distribution. Since this business landscape requires the musicians to act as managers, lawyers, and social media experts or to employ other people who have the knowledge and the experience to have these roles on their behalf [46], the artists should present an entrepreneurship attitude.

In the current digital environment that has changed music circulation and listening practices linking them with technology, new infrastructures as sociotechnical entities have emerged in many forms. An important issue in this new environment is the capacity for commodification; through platforms and mobile applications content may become tradable data [47] and new entrepreneurial opportunities arose. Mobile applications are a helpful tool that through visual and aesthetic interfaces have changed the rules in industries and markets mainly because of their free-based business models. Powerful mobile app-based business networks aid in improving the long-term efficiency of Small and Medium-Sized Enterprises (SMEs) [48]. One of their assets is that mobile applications do not always ask customers to pay for the service [49]. Another advantage for young entrepreneurs is the fact that acquiring knowledge of their designing techniques and their display information on different devices is a procedure that can be performed easily from anywhere and by anyone. Moreover, mobile applications are considered a strong branding platform for marketers [50].

Social distancing created by the COVID-19 health crisis has brought about major difficulties in all cultural industries [51]. Artists, trying to ensure their sustainability, were forced to find new ways in communicating with their audience, as festivals, concerts, and all kinds of live performances were canceled, adapting to the demands of the pandemic. Gradually, benefits started to come into businesses that invested in digital platforms [52].

In fact, a pandemic can work as an eye-opener for entrepreneurs that considered mobile apps to be optional [53]. Therefore, new strategies that would include technological developments were designed either by large institutions or by individuals themselves. The speed that drove those unprecedented challenges has brought respective changes in education as well to meet the demands of the market by providing knowledge and skills for creating entrepreneurially equipped students [54]. Mobile applications, when applied in the educational procedure, may provide a smooth blending across formal and informal learning environments as they engage learners with digital resources [55].

From a practical viewpoint, this fact drives potential entrepreneurs to have a positive predisposition to increasing their motivation to choose a new career path despite their age (most of them are very young) gender, education, and cultural background. Furthermore, it helps them carry out strategically cultural missions and face risks while creating a balance between managerial values and innovation [56]. Since entrepreneurship education is complex, both educational and social institutions should identify ways to earnestly support entrepreneurial creativity, inspiration, and innovation [57]. According to Cattani and Ferriani [58], entrepreneurs who have an interactive social presence are in a favorable position to achieve creative results working in a team. In addition, Chen and Chang [59], claim that networks indeed influence entrepreneurial success through the effects of available information through open access. Moreover, Mylonas and Petridou [60] claim, by the findings in their survey of female entrepreneurs, that venture performance in creative industries can be predicted by factors that affect conventional venture performance. According to the empirical analysis in their study, creative personality and professional network ties are regarded as the factors with the highest impact.

The current technological environment, as described above, that has been embraced by many actors, artists, users, and entrepreneurs comprise the reason that led the authors who, as academic researchers in the Universities of Macedonia and Aristotle University of Thessaloniki as well as in the University of Western Macedonia, Kastoria, in Greece to familiarize new entrepreneurs, who work in the music industry, with mobile applications in an effort to aid them in building their own professional identity. As the results that arise present the strengths, opportunities, and weaknesses of this effort, the educational procedure can be amended to equip them with the tools to marketize their work and evolve a financially sustainable career [61].

## 2. Materials and Methods

The methodology selected for the study was the expert interviews as the appropriate qualitative research method since the study involved 23 young entrepreneurial musicians, aged from 19 to 40 years old, who were selected after having attended a seminar on Management of Cultural Heritage, Communication, and Media. The experts' interviews are the most widely employed method of data collection in social sciences, as they provide an exceptional source for information retrieved from "inside" [62–65]. Since the participants relate to music either directly or indirectly as owners of music foundations, managers of musical events-concert halls, radio producers, owners of recording studios and music incubators, and of course as musical instrument players, they may contribute with their knowledge covering a great spectrum of the research area as experts. After having attended 2 two-hour fast-paced courses in creating cultural content mobile applications with software provided by Andromo, they were asked to design, implement, and upload their own applications which would promote:

1.  Various kinds of music (i.e., jazz, music from Epirus—a district in North-West Greece).
2.  Their Curriculum vitae emphasizes their artistic activities as well as their educational experience.
3.  Folk/traditional music from the local communities they originate.
4.  The creation of educational applications (how to learn to play a musical instrument).

The courses were carried out during the second wave of the COVID-19 pandemic in Greece, from 1–30 April 2021 as an educational section of their attendance of the seminar.

At this time, a new lockdown was imposed, and it was the reason for the cancellation of all musical activities that would take place live. The applications that were created ranged in a great spectrum which, in essence, presented the relative needs of the participants. Consequently, short descriptions will follow to make more comprehendible the framework they worked on along with their interests.

Some of the participants decided to promote their Curriculum Vitae either as musicians or as music teachers/instructors. These applications were enriched with their distinctions, samples of their artwork, and performances through photographs and certifications. Additionally, they used the capabilities offered to connect the application with other media, such as their official channel on YouTube, social media accounts, or other already created portals and sites in an effort to strengthen their profile and declare further their presence in innovative communication forms. Furthermore, additional information on upcoming concerts and activities was provided, while a very interesting effort was the attempt to add contact details for booking at events.

In a very impressive way, the theory and practice of various kinds of music and instruments (i.e., byzantine music, drums, folk instruments, such as bouzouki and political lute—laftas, etc.) were developed as music courses following detailed educational procedures. Another category of applications involved the promotion of a selected type of music, where tributes to jazz, traditional music of Epirus (a district in northern Greece), the island of Crete (Greece), and Cyprus were created. In these cases, definitions of the corresponding kind of music, its history, analysis of the instruments used, tributes to known artists, and even information and techniques on composing were included. In the same framework, contemporary music currents, referred to musical styles of China, Japan, and South Korea, aiming at younger audiences who want to explore new sounds and artists were unfolded.

Finally, the fact that many of the applications that were developed concerned the capability of new professional relationships with other musicians or with prospective students is of great importance. There was a great interest by the young entrepreneurs in attracting, approaching, and communicating with other parties for job opportunities via job employment and offering through CVs promotion. In one case, even fundraising was considered a possibility. Furthermore, a channel of communication with the audience was sought and collective promotions of a certain region's artists addressed to anyone interested were deployed, depicting the enhanced interest of the participants to reach the public.

Afterward, for the scopes of the study, following a case study framework, they were requested to take part in semi-structured interviews which ranged to open-ended conversations [62]. They were informed of the topics involved for a basis to be created prior to the conduct of the interviews at the level of the discursive consciousness [65]. However, they were urged to unravel their own perspectives and ideas freely, therefore prefixed guidelines and closed questions were avoided. The goal of the employed method was to assemble opinions on the way digital network platforms could be used to enforce musicians' entrepreneurial role, specifically on the relatively newly imposed practice of mobile applications. The participants, as people involved in a wide range of aspects of the music field, could be characterized as experts since they possess specialized knowledge obtained during their professional route, not only through a technical framework but also by engaging in organizational procedures [62]. This approach aims at receiving deeper insights into the current trends of digital network platforms employment for musicians' entrepreneurialism in Greece through a variety of acquired professional opinions.

Thus, following the application's implementation, individual interviews were performed in May 2021 through online presence. Initially, the applications were presented and the participants after being informed by the researcher that the procedure would be recorded, however, their anonymity would be protected, they provided their consent to start the interview and they were asked to answer the queries that succeed. It must be mentioned that these questions consist of the main axis of the interviews which were deployed even further as they ranged to open-ended conversations in a friendly environment to meet the needs of answering the RQs posed by the researchers. The aim of this case study was

to reflect upon the dynamics of new technologies in music entrepreneurship through the analysis of the young entrepreneurs' perspectives. Therefore, the questions were phrased:

1. To what extent do you think new technologies and specifically the applications of mobile devices/smartphones, such as the one that you created, are useful in the promotion of your project?
2. In what ways do you think mobile apps, such as the one you created, help promote your artwork and other music-related topics?
3. What are the elements that differentiate mobile applications, such as the one you created, from other non-linear technologies in the promotion of artwork and other music-related topics (YouTube channels, social media—Facebook/Instagram/LinkedIn, Spotify)?
4. What other form of communication (i.e., website, newsletter, social media accounts), besides the application that you created, would you choose to promote your project (linear or non-linear) and why?
5. Do you think that during the COVID-19 health crisis, mobile applications, in general, such as the one you created, worked successfully? If so, why? If not, what are the reasons that it did not work out as you expected?
6. According to Nielsen's 5Es, how do you evaluate your application and how do think your target audience does?

Regarding the last question of the interviews, the participants were asked to present their application to five representatives of their targeted group and asked them to evaluate it by answering whether it met the following criteria (5 Nielsen's) criteria, effectiveness, efficiency, engagement, ease of errors, ease of Navigation [25,66].

After completing the interview procedure, the collected material was transcribed based on the answers obtained from each question to aid in the process of interpretation, however, following the bibliography [65], the transcription was not detailed to each word. The first phase of the analysis of the expert interviews involved the evaluation of the context derived through the interviewees' statements regardless of the order of appearance. Of course, before exploring the transcribed texts from an interpretive point of view, it was simply read, to acquire the basic idea of each participant's interview [67]. During the evaluation phase, the authors' attention was centralized on the elements contained in the most intriguing answers. Any additionally raised comments and ideas, besides the initial scheduled, were noted, considered, and added to the outcomes even if they were of little relevance to the research questions [68,69]. Finally, the most interesting parts encountered were cross-examined to find out whether they could be formulated differently or incorporated into diverse categories, and consequently, a guide was created.

Consequently, the content of all interviews was analyzed following the principles of the thematic analysis [67,69–71]. Therefore, it was read carefully by two of the authors to enhance the reliability and credibility of the research and the initial codes appeared in vivo, that is they emerged from the original phrases of the participants [69]. For the analysis, all the steps mentioned in the literature were followed carefully since they were all performed by the authors, and no software was employed at any stage.

Afterward, based on the research questions, the establishment of codes in text sections was initiated, and consequently, a code book was created by each one of the authors that were involved in the process. Consequently, they worked together, realizing they arrived at similar conclusions and reflecting on scientifically revising codes when deemed necessary, to reach the final code book that was used in the analysis. Interpretations that were deduced during that procedure were "synthesized to form meta-inferences at the end of the study" as suggested by Teddlie and Tashakkori [72] (p. 20). In the next stage, thematic categories were constructed, based on descriptive data derived from the transcript interviews after organizing similarities and differences [73]. The formed categories were carefully revised to come to the final ones that would facilitate the aim of the research without being complex. In the following step, interpretive data acquired from the interviews, in general, were considered through the authors' critical perspective on the excerpts [71] and were added to the thematic categories which were classified in relation to the interviews questions as a

wider framework of the research questions to reaching the study's aim easier and initiating the interpretation of the emerged issues. As this procedure was completed, the authors concluded the six (6) themes that were regarded as the essence of the research, naming them accordingly while excerpts from the data were transferred to them to offer a clear understanding of each one of them [28]. The final stage was to form the academic report which involved discussion and interpretation of the data to explain the aspects of the study. The added interview excerpts were translated from Greek to English and the authors tried not to lose distinctions of the context.

## 3. Results

The presentation of the derived results will be made accordingly to the six main questions asked during the interviews, as thematic categories of the performed analysis, however, at the same time, the research questions posed for this study will be answered as well. The coding employed, e.g., I.1, stands for Interviewee no. 1 to certify the anonymity of the participants.

### 3.1. Usefulness of New Technologies in Promoting Musicians' Work

Most of the participants, regardless of the type of application they created, considered it as an important communication tool that is useful in terms of promoting their work. They were positively disposed to the employment of new technologies and specifically smartphone applications in publicizing their artistic creations. I.5 stated, "I believe that it achieved my initial goals. I agree that non-linear technologies enhance the promotion of artwork". In the same framework, I.7 explains that "[ . . . ] it succeeds to a great extent its purpose and in general new technologies reinforce this trend". One significant point made by the young entrepreneurs was that the applications are effective since they may easily reach an audience that is acquainted with these kinds of approaches. As I.10 points out, "It is provided free of charge or for a small fee to the public. New technologies are generally very useful nowadays, especially to an audience that is quite familiar with the Internet". Furthermore, they believe that these tools may even engage younger people in musical genres that are not extensively known or easy to comprehend. I.8 characteristically states, "I do not feel particularly familiar with new technologies, but I find that creating an application with artistic content attracts young audiences, especially to music genres that are difficult and less digestible as in the case of Byzantine music in which I specialize," I.11 further adds "My goal is the deepest and most essential rubbing of the Greek people with Jazz music. Through the acquisition of music-historical knowledge about it, they may appreciate it more as they will now know the historical context around which it was created and to strengthen the interest for the opening of more 'jazz clubs' in our country!" In the same framework, they think that even the personal presence in concert may become more interactive using technology. I.4 added further that "[ . . . ] non-linear technologies are useful. Specifically, the use of websites, social media, the creation of a YouTube channel, and the physical presence at concerts is more interactive with the audience we address as musicians". However, one participant expressed her apprehension on the issue, as she supposes that the applications are useful, however, they should be operated along with other forms of promotion; I.1 points out "Using a cultural application to learn a musical instrument can be ancillary rather than exclusive. What is available on the internet is not enough, lifelong learning is essential".

To conclude, as derived from their answers, the digital era in the music industry has opened the door for emerging artists who found new forms of communicating with their audiences. This is achieved through the employment of applications that aided them to acquire the exposure they deserve which otherwise would be difficult to do, especially with kinds of music that are not widely known. Advancements in music technology may spur entrepreneurial growth in more domains besides promoting solely music by establishing places like clubs where musicians may further benefit as entrepreneurs.

### 3.2. The Ways That Mobile Apps Help Promoting the Artwork

Consequently, the young entrepreneurs were asked to state their opinion regarding the ways that mobile applications could help promote their artwork. The most often appearing answer involved the contact with the audience. It is made obvious that the artists need to communicate their work and become known more broadly and new technologies may provide that, especially to the younger public. As I.8 states "I find that young people are more directly informed by using such an application. I believe that new technologies work more dynamically than traditional media, especially for the younger ages who are familiar with them". In the same perspective is the answer given by I.15, "I believe that designing and creating an application that is directly or indirectly related to music can evolve the form of communication and create new stronger ties with the young audience". Furthermore, it is pointed out that the inherent interactivity of smartphone applications "keeps the audience alert" (I.14) in a "more enjoyable" (I.17) "interesting" (I.16), and "*entertaining*" (I.5) way. Moreover, the immediacy that is offered through the employment of this kind of communication may bring the audience closer to the creators since "those interested would download it (the application) and be informed about the artist's participation in concerts and his artistic work" (I.2), and also, it "enhances the visibility and popularity of creators" (I.13). Lastly, but highly considered is the aspect that relates the use of new technologies with professionalism, since a self-created application may inform more concisely and earn respectability for the creator, "I believe that this digital application gives greater professionalism and is highly valued by my audience" (I.18).

The visibility and popularity of independent musicians may be enhanced using new technologies in an interesting and attractive way, especially when referred to the younger audience who are accustomed to using the current applications. The Internet made it possible for independent musicians, even those that present alternative forms of music, to build their careers on their own, without the intermediation of other corporations, as innovative approaches could be developed and new business models appear [18]. By receiving the appropriate knowledge on entrepreneurial management and on applications training related to their artistic work, they may be provided with the supplements to establish their identity to the audience and simultaneously present original professionalism to potential partnerships.

### 3.3. Elements That Differentiate Mobile Applications from Other Non-Linear Technologies in Promoting Artwork

The participants were asked to express their opinion on the differentiation of the mobile app they have created with other forms of non-linear technologies used for the promotion of music-related works, such as YouTube channels, social media, and Spotify. As they claimed, the application, if created correctly, may provide more and better-structured information that does not have an expiration date like the stories and polls on social media. The young entrepreneurs mentioned that although the application is not as easy as a blog to be updated, it finally may contain details in a classified way. However, they think that their combined use with other platforms may deliver the best results, "[ . . . ] the applications that work best are those that have access to the respective artist's YouTube channels" (I.5); "it subconsciously passes to the user that this artist has worked hard [ . . . ] for the way he communicates with his audience, if, of course, he has invested in a weekly contact with the public through social media" (I.19).

Mobile apps stood at the forefront of the continuing tech revolution in the industry in general. Nowadays, through a do-it-yourself iPhone, iPad, Android, and HTML5 app platform that allows the creation, editing, and publishing of artwork, musicians can proceed in the promotion of their work without any programming knowledge needed. As stated by themselves they can be recognizable and publicized within their genre in a way that is in accordance with their fans' options and still be free to follow an individually unique approach. Mobile applications use is not ephemeral; however, the social media approach is more extended. Therefore, to satisfy the audience's needs, combined employment of social

media platforms, such as Twitter, Instagram, and Facebook, under smartphone applications would result in achieving success in the commercial aspect of the music business.

### 3.4. Other Forms of Communication Selected for Promoting the Artwork

The question concerning other communication forms used for the promotion of the participants' work involved all kinds of publicity, such as traditional media (electronic and print), along with the new technologies, such as newsletters and social media. The answers were diverse and covered a great spectrum, both in the new technologies field (social media and websites) and in the traditional media as well. Regarding the new technologies, watching concerts through streaming services was pointed out as helpful to keep in touch with the audience along with social media posts. I.3 said, "Social media is more successful in attracting a young audience that is growing rapidly thanks to the tools offered to us (entrepreneurs)". I.4 continuing the former statement claimed, "I should add here that LinkedIn helps to find partners from abroad, while the other media mainly from inside". Creating partnerships is an issue that is important to them, and they search and find solutions through social media. I.15 answered this question "I am mainly in favor of non-linear technologies—modern forms of communication and social media accounts, especially LinkedIn. Cases of collaboration also arise through Instagram". There were also answers focusing on the employment of traditional ways, such as radio and newsletters distribution and this was principally justified as methods to approach older age groups. "The radio holds more dynamics and allows telephone communication. [ . . . ] and attracts older age groups," mentioned I.3, while I.12 added "[ . . . ] technologies based on traditional media, such as television and especially radio are just as important for the promotion of our work of art". I.8 further commented that "if one wants to promote their work to older people then they will follow the traditional methods of communication (newsletters, press releases, correspondence with invitations to friends of museums, places of culture, etc.), the posters have equally positive results". Another issue that seems to worry a lot of the young entrepreneurs is the cost of the publicities on various media "If you want to advertise on a cultural content website it costs very little compared to other ads in the press and on TV and even on local TV channels," stated I.22 and in the same framework, I.8 added "advertising through radio shows cost less to the creator compared to TV commercials. It is worth noting here that many young creators prefer to pay for online advertising as it is a more economical solution than promoting them through traditional media".

It seems that the publicity cost is an issue of great importance for young entrepreneurs, and it is mentioned greatly. They want to communicate with their audience, and they are anxious about this process. Although they use novel technologies, they still believe in traditional ways of approaching the fans, such as posters and marketing through TV and radio appearances, probably as a need to satisfy the current audiences that are accustomed to this form of communication with the artists.

### 3.5. COVID-19 Health Crisis and Mobile Applications

The difficult era that we are going through due to the COVID-19 pandemic could not be undiscussed, especially since the creation of the applications and the conduct of the interviews took place during the lockdown in Greece. Therefore, one question concerned the successful—or not—performance of the mobile applications during the health crisis and the relative reasons for both cases. The participants when expressing their opinion were positively disposed to the use of the applications and new technologies since it was a way for the artists to keep their audience active. As I.10 stated, "During this period, just because all the concerts were canceled, people needed something to listen to. They had more time to spend looking for their favorite artists and "different" songs. That's why they followed artists, music genres, downloaded relevant applications in an effort to stay active and up to date with new trends". In general, they think that through this type of promotional action quality content was projected in a more modern way. An issue raised was the content of the applications since in order to be successful, the young entrepreneurs thought

that it had to include information on the activities they performed. Characteristically, I.1 pointed out "New technologies have certainly enhanced the promotion of contemporary artwork. [ . . . ] The cultural application in my case did not help so much in the promotion of the artistic project. But this is because I chose a topic related to the musical instrument that I deal with, but I did not include in the application further information about the actions in which I participated". As expressed in their answers, they think that this form of communication will continue to exist even after the health crisis, especially in the cases of remote areas that are deprived of access to live events. An interesting conclusion emerged from the fact that the participants followed the actions of important institutions related to the music industry and based on their examples, they think that these applications will work positively. However, at this point, it has to be cited that many of the participants expressed their aspect that nothing is compared with live communication. As I.22 stated, "Unfortunately, the dynamics of a concert cannot be revealed on social media. You cannot awaken the various emotions that are created when you listen to a concert live".

In an attempt to provide enriched cultural products during the COVID-19, artists tried to find alternative ways to communicate with the audiences, while in parallel keeping engaged all those who were in need of acquiring cultural events. Evidence shows that the way people listen to music is changing due to the coronavirus. Changes to listening behavior during the pandemic, with more consumers using home applications on TVs and smart devices were reported. Under this spectrum, the use of mobile applications during the COVID-19 period was increased by entrepreneurs as a means to promote their work [74], and artists that possessed the entrepreneurial identity were able to find solutions during the crisis [75].

### 3.6. Application Rating According to Nielsen's 5Es

Finally, the young entrepreneurs were asked to rate according to Nielsen's 5Es (efficiency, effectiveness, error resistance, easy navigation, and engagement) the created application themselves and their audience as well. Most of the answers were very positive on the first four factors while many of the participants stated that, as designed, there was not a lot of engagement in the application. Regarding the efficiency factor, to summarize the replies, the applications were flexible, they presented simple operations so that even an inexperienced user could navigate themself. The effectiveness was also rated highly since according to the justifications, the users could be sufficiently informed through the information provided and the design could contribute to a pleasant experience. In relation to the error resistance factor, they believe that errors are prevented through the immediate and easy organization of sections and the clear terminology of the content, nonetheless, in some cases errors surfaced. The participants realized their mistakes and are willing to correct them in a new effort. The easy navigation factor was also highly rated due to the easy and flexible section organization and the lack of difficulty in returning to the first page. Regarding the engagement element, they realized its drawback during the trial period. They found out that it had to be linked with social media accounts, other sites where the artist is referred (concerts, or achievements) or a section of comments to be inserted. The only engaging feature, in some of the cases, was the capability of sending an email. An important comment made by I.19 is "[ . . . ] what most agreed on is that it provides me with a professionalism, and it is a digital agenda, in which my fans can look back on my participation in various activities".

Once the audience is captivated by an application that may be spread out via social media, personal blogs, or any other communication form, then it is easier to be engaged and return with anticipation to find information on the artist and his/her latest work. This is a strategy that needs to be followed in order for a successful course as an entrepreneurial artist to be achieved, in the sense of adding new audience members but also providing an enhanced experience while developing ongoing relationships [66,76]. Therefore, testing the factors that measure the application's effectiveness is a crucial procedure that should be performed.

## 4. Discussion

Novel technology and cultural changes due to economical and health crises have affected the music industry. The emergence of new technology trends, including digital tools or infrastructure, digital platforms, such as websites, blogs, and mobile applications, along with the presence of social media has transformed the way music is consumed and produced. Artists-entrepreneurs with small and larger music businesses, cultural organizations, and educational institutions have tried to adapt or transform to the changes of times. Entrepreneurship combined with creative skills and the ability to imagine and identify opportunities and schedule a professional course can finally build sustainable development.

The traditional way of promoting a music business in the music industry presented limited opportunities to the artists that would prefer to market themselves while the use of digital communications technologies may provide varied marketing strategies, therefore reaching varied audiences and potential collaborators [76–78]. Access to distribution channels was not easy and artists required help from professionals to promote their work or businesses [79,80]. However, nowadays, digital technologies are vital components that play a significant role in mediating deeper relationships between artists and audiences, and new digitized business models arise in the arts and cultural industries [81]. In another aspect, progress in recording, managing, and authoring new narratives may benefit average users who may become active participants in the processes of experiencing and evaluating new forms of content through the associated services [24]. This article aimed to present an approach to the contribution of mobile applications in the promotion of entrepreneurs related either directly or indirectly to the music industry. Upon completion of the design, creation, and uploading of their mobile applications they agreed on the following common points. The first one is that they consider that using mobile applications for their entrepreneurial work is a modern communication tool, which directly reaches the public to which they are addressed. In another aspect, most of them chose in their application to include information from their Curriculum Vitae, their participation in artistic activities, and the promotion of their teaching experience. A few of them recognized that the import of the corresponding links to their YouTube channel, their official website, and social media, is an easy and quite creative procedure. Unfortunately, as they lack experience in the creation of this kind of application, some of the links that they added did not lead to the corresponding pages. A significant percentage of them expressed the viewpoint that with constant updates in their application, they achieve the corresponding interaction they want with their target audience. Moreover, they consider new media (nonlinear technologies) as important; in order to keep their audience active, but also out of the traditional media (linear technologies) they consider radio as an equally important medium. Finally, they support that the applications in combination with social media and their official websites can create their corporate-entrepreneurship identity (identity of themselves as creators or as the owners or employees of a music company). Most of the applications (20/23), based on the evaluations performed by the entrepreneurs with the selected audience, were efficient and effective, easy to navigate, and did not show any particular errors but only some of them (3/23) solved the encountered difficulty in matching links with their respective channels on their social media. Most of the participants expressed the opinion that their application could be more interactive, however, this was feasible since they were not well acquainted with the Andromo environment, although during the courses they learn ways to achieve interactivity, a fact which supports that more practice is required. Analyzing further the results of the case study and answering the RQs posed by the authors, the following interpretive conclusions were made. Regarding RQ1: Are new technologies effective in promoting the work of young entrepreneurs in the music industry? we could argue that according to the answers received, new technologies are effective for that purpose. The participants consider that using mobile applications for their entrepreneurial work is a modern communication tool, which directly reaches the audience they address to. Especially, when referring to younger ages, they believe that it is of great importance to encompass these techniques. Through the process of designing

the application, their creativity was stimulated to conceive innovative ways of introducing themselves and their work. Furthermore, they think that they empowered their presence through digital communication tools which worked in favor of their professionalism as they could match demand and offer [82]. Mobile applications, through the appearance of brand awareness, are efficient in increasing brand equity [50]; thus, artists may become known through smartphone applications and enhance their visibility and stabilize their position as entrepreneurs.

Entrepreneurs related to the music industry who participated in the present study, exploited the new technologies and reached successful results, as stated themselves after having evaluated their applications. Most of them chose to include information from their Curriculum Vitae, their participation in artistic activities, and their teaching experience, to be further self-presented. Additionally, the corresponding links to their YouTube channel, official website, or social media worked effectively in expanding their publicity. Therefore, through our findings, the answer to RQ2: Are smartphone cultural applications helpful for entrepreneurs related to music? is extensively affirmative. According to them, the introduction of these kinds of applications along with social media networks (Facebook, Instagram) and streaming services (YouTube, Spotify, Apple Music), are holding a crucial role in promoting artists and artwork, reaching a wider audience bearing sustainability in their businesses as they help in building a corporate-entrepreneurship identity (identity of themselves as creators or as the owners or employees of a music company). A corporate identity that is characterized by immediacy and interactivity with the audience and may present the form of a digital album-diary that everyone can look at any time he/she wants from everywhere in the world is a significant asset [83].

As explained above, our RQ3: Were web communication technologies favored by young entrepreneurs in publicizing their work? is mainly answered. Nevertheless, audience engagement is an essential issue, nowadays, and is recognizable by entrepreneurs who want to have an effective relationship with their followers. Therefore, they realize the need to develop other skills rather than just performing/composing music, (promoting, distributing, brand development, and having financial knowledge). However, they need more practice time to learn how to embed engagement features in their applications to effectively accomplish this demand. Contemporary entrepreneurial methods should be adopted to help in the evolution of their career and build a fan base. An interactive mobile application that is linked with formal accounts of the musicians' social media, could lead consumers to additional entrepreneurial actions, such as consuming and buying their music and even buying concert tickets.

Despite that, it has to be mentioned that traditional methods of reaching the audience, such as radio and television commercials and posters, were also considered important, especially, as compelling approaches to older people. This enhances the entrepreneurial perspective as they care about the spectrum of their potential audience.

Regarding the last research question, RQ4: During the COVID-19 period, were the communication technologies thought of as helpful in promoting their artwork by the young entrepreneurs in the music field? according to the findings, mobile applications, websites, and social media were the most important technologies used in music promotion during this turbulent period. As the imposed lockdowns prohibited in-person concerts, the audience needed a new communication pattern with their favorite artists, and it was found in the use of mobile devices since their primary purpose is connectivity that provides new affordances in communication [55]. Thus, the fans followed those who were active on social media or other platforms and could be reached primarily via mobile devices. The participants realized that they had to include a lot of information regarding their work and especially the projects they worked on to be successful in this contemporary genre of communication. Although they think that this form of connection with the audience will continue to exist, as artists that need close conduct with people, they support that the dynamics of live concerts cannot be compensated. As other research supports, the artists

that possessed entrepreneurial identity during the crisis were able to find solutions and manage the process of change [75].

To conclude with, as the Internet and digital technologies, in general, have become essential parts of communication between artists and audiences [18], the young music entrepreneurs, by adopting a knowledge-based perspective on the creation of a mobile application that will facilitate the transfer of their professional identity and the promotion of their artwork, will be helped in initiating new ways of starting a new venture. Once their idea is validated and enriched through the employment of digital technologies, then it may be promoted via marketing strategies and further distributed to their potential audiences while establishing their unique place in the globalized cultural market [42].

## 5. Conclusions

This article aimed to present an approach to the contribution of mobile applications in the promotion of entrepreneurs related either directly or indirectly to the music industry. The emergence of virtual networks for entrepreneurship promotions leads people who are related to music to use innovative solutions reshaping old forms to adjust to the new platformed media environment. Designing, implementing, and publicizing their own mobile applications—since everyone nowadays uses their mobile phone to download useful applications—is a way to effectively follow the contemporary demands of marketing. Through this practice of mobile marketing, the creation of a brand that can be stored in the memory of prospective clients/audience—who afterward can become loyal fans—may be aided [50]. In this way, they approach and expand their target group and possibly receive bigger earnings. It is in the authors' belief that using these findings the academic society will be aided in the way mobile applications can be useful tools for entrepreneurs' cultural promotion as well as means to facilitate the audience's engagement with different music genres.

Current technologies have made it possible for musicians to produce, promote and distribute their music through meaningful communication channels without the intermediation of a brand [18]. The findings of the present study exhibit the effective relationship between music entrepreneurs with new technologies, as other relative studies on smartphone capabilities with respect to the cultural sector point out as well [84,85]. During the COVID-19 health crisis, entrepreneurs showed a lack of optimism and insecurity about their work. Many lockdowns were forced and that was the reason for the cancellation of all the live performances, artistic events, and musical concerts. The creation of mobile applications was linked to their psychological empowerment and approach of a new perspective in the promotion of music against the feeling of inactivity caused by these cancellations. Design and the development of applications related to music were significant tools for entrepreneurs to maintain a healthy relationship with music lovers. As these applications were deployed through a pedagogical procedure, the participants were not reluctant in proceeding to their creation. The educational environment provided the motivation to overcome any obstacles explore new concepts and expand their creativity and entrepreneurial aspects. As with all studies, this one too comes with limitations, one of which is the small number of participants in the research which does not permit the generalization of the results; however, it provides a clear insight on the subject. While accomplishing the project, varied individual factors were indicated which could be used as starting points in succeeding research designs. In future work, it would be interesting to reach more young music entrepreneurs as well as artists from other cultural domains in quantitative research to acquire more global results. Furthermore, the audience's perspectives would provide feedback on many terms, such as the Nielsen 5 Es, and shed light on the potential value of the applications through the users' prism.

**Author Contributions:** Conceptualization O.P.; Data curation, O.P., M.M. and C.A.D.; Investigation, O.P.; Methodology, M.M.; writing—original draft preparation, O.P. and M.M.; writing—review and editing, M.M., C.A.D. and G.M.K.; Supervision, M.M. and G.M.K. All authors have read and agreed to the published version of the manuscript.

**Funding:** This research received no external funding.

**Institutional Review Board Statement:** Ethical review and approval were waived for this study, since all participants were adults that provided informed consent prior to their participation in the research.

**Informed Consent Statement:** Informed consent was obtained from all subjects involved in the study.

**Data Availability Statement:** Data are available upon request to the authors.

**Conflicts of Interest:** The authors declare no conflict of interest.

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
