# Peer review of "The Significance of Digital Network Platforms to Enforce Musicians’ Entrepreneurial Role: Assessing Musicians’ Satisfaction in Using Mobile Applications"

_sustainability, doi:10.3390/su14105975_

Round 1
Reviewer 1 Report
This paper presents an academic and social interest topic about mobile applications for musicians. This works deals with a current topic because technology is being implemented in all areas of knowledge and arts, including music.
The manuscript, which continues the adequate structure of all academic work, first presents a bibliographic review based on the most innovative specialized literature. Later, a quantitative study is presented, whose application and results can serve as inspiration to other contexts. No methodological errors are observed. However, the sample is small, as authors themselves affirm in the conclusions section. In this regard, I believe that the paper should insist that it is a case study.
A review of final conclusions is recommended because we can observe that a sentence is not finished (see p. 12, “Another limitation concerns the”). A review of final bibliography is also recommended to adjust some references according to the journal normative.
For all these reasons, I recommend that the article could be accepted after minor revision.
Author Response
This paper presents an academic and social interest topic about mobile applications for musicians. This works deals with a current topic because technology is being implemented in all areas of knowledge and arts, including music.
The manuscript, which continues the adequate structure of all academic work, first presents a bibliographic review based on the most innovative specialized literature. Later, a quantitative study is presented, whose application and results can serve as inspiration to other contexts. No methodological errors are observed. However, the sample is small, as authors themselves affirm in the conclusions section. In this regard, I believe that the paper should insist that it is a case study.
Answer: The authors would like to thank the reviewer for these motivating words which make them keep working hard. They have considered the reviewer’s suggestion and in the revised manuscript they refer to the research as a case study both in the abstract as well as in the main text.
(Page 1, lin. 16-22). In this framework, entrepreneurs-musicians who attended a seminar on Management of Cultural Heritage, Communication and Media in Greece were asked to promote their activities through the creation of a smartphone application. After having completed their endeavor and further evaluated it, they participated in a qualitative research based on the theory of experts’ interviews, as a case study. The aim of this paper, through thematic analysis of the conducted interviews, is to reflect upon the dynamics of new technologies on music entrepreneurship.
(Page 2, lin. 96). In this framework, the present investigation, as a case study, attempted to answer in a prime stage, whether mobile applications hold a significant role in entrepreneurs’ promotion through an educational, though addressed to professionals, process.
(Page 6, lin. 260). Afterwards, for the scopes of the study, following a case study framework, they were requested to take part in semi-structured interviews which ranged to open-ended conversations [46].
(Page 6, lin. 282). The aim of this case study was to reflect upon the dynamics of new technologies on music entrepreneurship through the analysis of the young entrepreneurs’ perspectives.
(Page 12, lin. 614). Analysing further the results of the case study and answering the RQs posed by the authors, the following interpretive conclusions were made.
A review of final conclusions is recommended because we can observe that a sentence is not finished (see p. 12, “Another limitation concerns the”). A review of final bibliography is also recommended to adjust some references according to the journal normative.
Answer:The reviewer is totally right regarding the unfinished phrase which was an authors’ mistake. In the revised manuscript the conclusions section, as well as the whole of the text, was proof-read carefully. Furthermore, the references were adjusted to the journal normative.
For all these reasons, I recommend that the article could be accepted after minor revision.
Please see the attachment

Reviewer 2 Report
The organization of sections is satisfactory. The paper is of appropriate length and suggestions and conclusions are well-written.In the article the authors present the chosen method - interview. Unfortunately, it was not indicated on what basis the research grip was chosen. I would like to ask for an explanation in the article as it is quite important what factors decided about it.
The paper provides an overview is the dynamics of new technologies on music entrepreneurship on research. This paper should be of various researchers’ interest. The abstract is a correct. Authors indicate very clearly the purpose of the article. The key words are well written. The quality of the language is appropriate, however, the authors sometimes uses expressions in informal style and a few sentences are unclear to the reader. An in-depth literature analysis and research methods were used. It is worth elaborating on the topic in the future. The organization of sections is no satisfactory.
Author Response
Reviewer: 2
The organization of sections is satisfactory. The paper is of appropriate length and suggestions and conclusions are well-written.In the article the authors present the chosen method - interview. Unfortunately, it was not indicated on what basis the research grip was chosen. I would like to ask for an explanation in the article as it is quite important what factors decided about it.
Answer:We are thankful for the positive feedback of the reviewer along with the respective constructive comments, which were taken into consideration in the revised manuscript. The motivation of this work was added in the Introduction section along with further explanation on the authors’ original thoughts.
(Page 2, lin. 80-95). The authors’ academic research interest lays among technology (sound, smartphone applications, etc.) culture (in the domain of cultural heritage as well) and technology-enhanced education [19-25] and comprehend that combined technological developments and entrepreneurial skills are important factors in modern economies. Under the prism of the contemporary social and business environment where the adoption of smartphone applications, which are becoming simple and natural to use, may lead to successful entrepreneurial endeavors, even when created by amateurs though under proper educational training and guidance, the authors were engaged in this research. The motivation of this work was initiated by the noticed urge among young people to find a way to express their creativity in innovative ways as entrepreneurs in a mature field such as the mobile applications to create audience awareness and engagement. Through their unique perspectivity regarding problem-solving of the activities of their own work after being challenged to think their needs in relation to their audience and prospective customers, and the feedback they may offer, contemporary forms of education could transform as to be focused to reach a higher level of successful training.
The paper provides an overview is the dynamics of new technologies on music entrepreneurship on research. This paper should be of various researchers’ interest. The abstract is a correct. Authors indicate very clearly the purpose of the article. The key words are well written. The quality of the language is appropriate, however, the authors sometimes uses expressions in informal style and a few sentences are unclear to the reader. An in-depth literature analysis and research methods were used. It is worth elaborating on the topic in the future. The organization of sections is no satisfactory.
Answer:The authors would like to thank the reviewer for pointing out these issues. In the revised manuscript the whole of the text was proof-read carefully, and confusing points were clarified.
Please see the attachment.

Reviewer 3 Report
Abstract:
The aim of the paper presented is poor: " to reveal the point of view of participants in a seminar?" It does not seem to be a scientific approach... Even if the paper is well structured and organized, in my opinion, makes no sense to present a paper whose main goal is the presentation of a point of view of a group of people!
This aim is not aligned with the RQ presented in the introduction. I suggest the authors to rethink the paper's main aim presented in the abstract.
Considering the 4 RQ presented in the introduction they should be supported by the literature review.
The literature review covers the essentials. Materials and Methods describe the procedures adopted, however, the relation between RQ (from the introduction and the interview questions (from methods and materials is not clear). Some of the latter questions seem to be a direct RQ, while some others don't.
Results
The results section presents an analysis of the questions presented previously. However, there is no content analysis, nor a qualitative one. It seems that the conclusions are based in authors subjective analysis.
Discussion
The discussion makes sense, but once again is based on author's subjective opinion.
The paper is interesting but it presents some scientific flaws:
- Main aim presented in the abstract makes no sense from a scientific perspective
- The RQ are not supported by the literature
- It is not clear how RQ were adressed
- Interview questions analysis is subjective
For the reasons presented above, I think that this paper needs a deep change to be published as a scientific paper.
Author Response
Abstract:
The aim of the paper presented is poor: " to reveal the point of view of participants in a seminar?" It does not seem to be a scientific approach... Even if the paper is well structured and organized, in my opinion, makes no sense to present a paper whose main goal is the presentation of a point of view of a group of people!
Answer:The authors would like to thank the reviewer for pointing out this issue. It was a mistake to present the research in such a manner since the purpose was more extensive, therefore, considering the reviewer’s comment it has been changed both in the abstract and in the respective parts of the manuscript.
(Page 1) Abstract: Digital entrepreneurship though the employment of smartphones and other handheld devices applications is an innovative form of customer approach. Particularly, in the cultural marketing sector, new technologies, such as social media, YouTube channels and mobile applications may increase the artists’ visibility and attract new partnerships and audience. In this framework, entrepreneurs-musicians who attended a seminar on Management of Cultural Heritage, Communication and Media in Greece were asked to promote their activities through the creation of a smartphone application. After having completed their endeavor and further evaluated it, they participated in a qualitative research based on the theory of experts’ interviews, as a case study. The aim of this paper, through thematic analysis of the conducted interviews, is to reflect upon the dynamics of new technologies on music entrepreneurship. As derived by the analysis, the use of mobile applications may effectively approach prospective partnerships and audience, especially if combined with other contemporary forms of communication and results in presenting enhanced professionalism. Audience engagement, which is an issue that is sought while attempting to optimize promotion processes, may be achieved if further practice is performed. As the research was conducted during the COVID-19 pandemic, the demand for this form of making publicly known their artwork was considered essential.
This aim is not aligned with the RQ presented in the introduction. I suggest the authors to rethink the paper's main aim presented in the abstract.
Answer: The authors have reconsidered the expressions used in the abstract as well in the main manuscript and they have adjusted both parts and in the revised manuscript they have added more and further explanatory context in accordance with their work such as the paragraph that follows.
(Page 2, lin. 80-95). The authors’ academic research interest lays among technology (sound, smartphone applications, etc.) culture (in the domain of cultural heritage as well) and technology-enhanced education [19-25] and comprehend that combined technological developments and entrepreneurial skills are important factors in modern economies. Under the prism of the contemporary social and business environment where the adoption of smartphone applications, which are becoming simple and natural to use, may lead to successful entrepreneurial endeavors, even when created by amateurs though under proper educational training and guidance, the authors were engaged in this research. The motivation of this work was initiated by the noticed urge among young people to find a way to express their creativity in innovative ways as entrepreneurs in a mature field such as the mobile applications to create audience awareness and engagement. Through their unique perspectivity regarding problem-solving of the activities of their own work after being challenged to think their needs in relation to their audience and prospective customers, and the feedback they may offer, contemporary forms of education could transform as to be focused to reach a higher level of successful training.
Considering the 4 RQ presented in the introduction they should be supported by the literature review.
Answer:Further studies were considered, and the literature review section was enriched to clarify the scope of the research. New additions may be found in many parts of the revised manuscript, such as the following:
(Page 2). As Pietila (2019) and Waldfoel (2021) mention, musicians – entrepreneurs became truly independent since digital technologies help them to enter the music industry easier in comparison to the past and provide them a variety of tools, especially the mobile applications [10,11]. By employing contemporary applications, Kalliris, Dimoulas and Matsiola (2019) claim that many processes which are related to music can nowadays be performed in digital home studios [12]. Music entrepreneurs can handle hardware with the use of music applications and that fact led to the emergence of a wider spectrum of talents because of the reduced expenditures. Traditional barriers of cost and skills have been removed, according to Dewenter et al. and Karubian [13,14] by technological developments, as direct exchanges between producers, who can provide their products while generating revenue, and buyers, who are benefited from alternative options and lower prices, are supported [15]. Musicians, nowadays, may earn retribution (financial and recognizability) easier than at any other era. Specifically, their fans, via their handle head devices, such as mobile phone, through registration to ap-plications or platforms such as Apple iTunes, may pay for the musical creations. Moreover, musicians also gain international perceptibility without geographical boundaries [16].
The literature review covers the essentials. Materials and Methods describe the procedures adopted, however, the relation between RQ (from the introduction and the interview questions (from methods and materials is not clear). Some of the latter questions seem to be a direct RQ, while some others don't.
(Page 4). Social distancing created by the COVID-19 health crisis has brought about major difficulties in all cultural industries [42]. Artists, trying to ensure their sustainability, were forced to find new ways in communicating with their audience, as festivals, con-certs and all kinds of live performances were canceled, adapting to the demands of the pandemic. Gradually, benefits started to come in to businesses that invested to digital platforms [43]. In fact, the pandemic can work as an eye-opener for entrepreneurs that considered mobile apps to be optional [44]. Therefore, new strategies that would include technological developments were designed either by large institutions or by individual themselves. The speed that drove those unprecedented challenges has brought respective changes in the education as well to meet the demands of the market by providing knowledge and skills for creating entrepreneurially equipped students [45]. Mobile applications, when applied in the educational procedure, may provide a smooth blending across formal and informal learning environments as they engage learners with digital resources [46].
Answer:The authors would like to thank the reviewer for providing the option to clarify the way the analysis was performed. In the revised manuscript, two of the RQs were rephrased to direct questions and more details were added in the description of the methods employed.
(Page 3). RQ1: Are new technologies effective in promoting the work of young entrepreneurs in the music industry?
RQ2: Are smartphone cultural applications helpful for entrepreneurs related to music?
RQ3: Are nonlinear communication technologies favored would by young entrepreneurs choose in publicizing their work?
RQ4: During the Covid – 19 period, were the communication technologies thought of as helpful in promoting their artwork by the young entrepreneurs in the music field?
(Page 6). Thus, following the applications implementation, individual interviews were performed in May 2021 through online presence. Initially, the applications were presented and the participants after being informed by the researcher that the procedure would be recorded, they provided their consent to start the interview and they were asked to answer the queries that succeed. It must be mentioned that these questions consist of the main axis of the interviews which were deployed even further as they ranged to open-ended conversations in a friendly environment to meet the needs of answering the RQs posed by the researchers. The aim of this case study was to reflect upon the dynamics of new technologies on music entrepreneurship through the analysis of the young entrepreneurs’ perspectives. Therefore, the questions were phrased:
- To what extent do you think new technologies and specifically the applications of mobile devices / smartphones, such as the one that you created, are useful in the pro-motion of your project?
- In what ways do you think mobile apps, such as the one you created, help promoting your artwork and other music-related topics?
- What are the elements that differentiate mobile applications, such as the one you created, from other non-linear technologies in the promotion of artwork and other music-related topics (YouTube channels, social media –Facebook / Instagram / LinkedIn, Spotify)?
- What other form of communication (i.e., website, newsletter, social media accounts), besides the application that you created, would you choose to promote your project (linear or non-linear) and why?
- Do you think that during the Covid -19 health crisis the mobile applications, in general and such as the one you created, worked successfully? If so, why? If not, what are the reasons that did not work out as you expected?
- According to Nielsen's 5Es, how do you evaluate your application and how do think your target audience does?
Results
The results section presents an analysis of the questions presented previously. However, there is no content analysis, nor a qualitative one. It seems that the conclusions are based in authors subjective analysis.
Answer:In the same framework of the previous comment, the authors, in the revised manuscript, present in more details the way they performed the thematic analysis of the conducted interviews.
(Pages 6 and 7). After completing the interviews procedure, the collected material was transcribed based on the answers obtained from each question to aid in the process of interpretation, however, following the bibliography [47], the transcription was not detailed to each word. The first phase of the analysis of the expert interviews involved the evaluation of the context derived through the interviewees’ statements regardless of the order of appearance. Of course, before exploring the transcribed texts from an interpretive point of view, it was simply read, to acquire the basic idea of each participant’s inter-view [50]. During the evaluation phase, the authors’ attention was centralized on the elements contained in the most intriguing answers. Any additionally raised comments and ideas, besides the initial scheduled, were noted, considered, and added in the out-comes even if they were of little relevance to the research questions [51]. Finally, the most interesting parts encountered were cross-examined to find out whether they could be formulated differently or incorporated to diverse categories and consequently a guide was created.
Consequently, the content of all interviews was analysed following the principles of the thematic analysis [52]. Therefore, it was read carefully by two of the authors to enhance the reliability of the research and the initial codes appeared in vivo [51].
Afterwards, based on the research questions, the establishment of codes in text sections initiated, and consequently a code book was created. At the next stage thematic categories were constructed, initially based on descriptive data which derived from the transcript interviews. In a following step, interpretive data acquired from the interviews in general were considered through the authors’ critical perspective on the excerpts [50] and were added in the thematic categories. As this procedure was completed, the thematic categories were classified in relation to the research questions to reach easier the study’s aim and the interpretation of the emerged issues initiated.
Discussion
The discussion makes sense, but once again is based on author's subjective opinion.
Answer:In the discussion section, more references of other studies were added to help in the orientation of our study in the research field (pages 12-18).
The paper is interesting but it presents some scientific flaws:
- Main aim presented in the abstract makes no sense from a scientific perspective
- The RQ are not supported by the literature
- It is not clear how RQ were adressed
- Interview questions analysis is subjective
For the reasons presented above, I think that this paper needs a deep change to be published as a scientific paper.
Answer:The authors would like to thank the reviewer for his/her constructive comments which were all considered in the revised manuscript.
Please see the attachment.

Round 2
Reviewer 3 Report
Dear authors, please find below some comments:
- Abstract:
"Digital entrepreneurship though the employment of smartphones and other handheld devices applications is an innovative form of customer approach." - though or through?
The abstract was improved from the previous version
- Research Questions:
I recognize that there were an improvement in the literature review, however the relation among the literature review and the RQ is not yet clear. The paper should provide and clearly state the theorectical support for each research question. Currently, there is a general (improved) literature review but is not clear what are the scientific works that lead to each RQ.
One of my previous comments:
"The literature review covers the essentials. Materials and Methods describe the procedures adopted, however, the relation between RQ (from the introduction and the interview questions (from methods and materials is not clear). Some of the latter questions seem to be a direct RQ, while some others don't"
After reading the new version, that presents some changes, I consider that the problem remains. As previously mentioned is not clear the state of art support for the RQ. As well it is not clear the support for the questions? Some questions were apparently designed to reply to a specific RQ.
- Used Methods:
The authors added a new sentence:
"Consequently, the content of all interviews was analysed following the principles of the thematic analysis [60]. Therefore, it was read carefully by two of the authors to enhance the reliability of the research and the initial codes appeared in vivo [59]"
This is not clear to me. The content analysis was done by 2 authors? Was any software used? what do you mean by "....codes appeared in vivo"?
The methodoly used is not clear. The performed analysis still seems that might be subjective.
Even with some changes and improvements from the previous version, I think that this paper still needs to be improved at the scientific aspect, in order to be published.
Author Response
Reviewer: 3
The authors would like to thank the reviewer for his/her interest in their manuscript. They have considered the reviewer’s comments and in the revised manuscript they have tried to address them all.
Dear authors, please find below some comments:
- Abstract:
"Digital entrepreneurship though the employment of smartphones and other handheld devices applications is an innovative form of customer approach." - though or through?
The abstract was improved from the previous version
Answer: The reviewer’s question is totally right. It was a spelling mistake; it was meant to be “through the employment”. The authors would like to thank the reviewer for his/her comment regarding the improvement of the previous version of the abstract
- Research Questions:
I recognize that there were an improvement in the literature review, however the relation among the literature review and the RQ is not yet clear. The paper should provide and clearly state the theorectical support for each research question. Currently, there is a general (improved) literature review but is not clear what are the scientific works that lead to each RQ.
One of my previous comments:
"The literature review covers the essentials. Materials and Methods describe the procedures adopted, however, the relation between RQ (from the introduction and the interview questions (from methods and materials is not clear). Some of the latter questions seem to be a direct RQ, while some others don't"
After reading the new version, that presents some changes, I consider that the problem remains. As previously mentioned is not clear the state of art support for the RQ. As well it is not clear the support for the questions? Some questions were apparently designed to reply to a specific RQ.
Answer: The authors are thankful once more for the constructive remarks and suggestions, which were further taken into consideration in the revised manuscript. More studies were used to support the RQs and clarify the purpose of the research (pages 1-5). However, we are not sure that we clearly understand the reviewer’s comment regarding the connection of the RQs and the questions posed to the participants. The interviews were centralized on the RQs however they had to be more extensive to cover a wider area and offer the participants the ability to express themselves more freely and perhaps other interesting conclusion would derive. The RQs are firmly stated using less words in an effort to be precise.
- Used Methods:
The authors added a new sentence:
"Consequently, the content of all interviews was analysed following the principles of the thematic analysis [60]. Therefore, it was read carefully by two of the authors to enhance the reliability of the research and the initial codes appeared in vivo [59]"
This is not clear to me. The content analysis was done by 2 authors? Was any software used? what do you mean by "....codes appeared in vivo"?
The methodoly used is not clear. The performed analysis still seems that might be subjective.
Answer: The authors would like to clarify that the thematic analysis was performed by them following all the steps that are indicated by the literature review. It was not used any software; however, all the stages in the conducted analysis were based on international literature processes. The methodology should be described more extensively from the beginning, instead of, erroneously, presenting it briefly. We hope that the descriptions that were further added are satisfactory.
(Page 8). Consequently, the content of all interviews was analysed following the principles of the thematic analysis [67, 69-71]. Therefore, it was read carefully by two of the authors to enhance the reliability and credibility of the research and the initial codes appeared in vivo, that is they emerged from the original phrases of the participants [69]. For the analysis, all the steps mentioned in the literature were followed carefully since they were all performed by the authors and no software was employed at any stage.
Afterwards, based on the research questions, the establishment of codes in text sections initiated, and consequently a code book was created by each one of the authors that were involved in the process. Consequently, they worked together, realizing they arrived at similar conclusions and reflecting scientifically revising codes when it deemed necessary, to reach at the final code book that was used in the analysis. Inter-pretations that were deduced during that procedure were “synthesized to form me-ta-inferences at the end of the study” as suggested by Teddlie and Tashakkori [72] (p.20). At the next stage thematic categories were constructed, initially based on descriptive data which derived from the transcript interviews after organizing similarities and differences [73]. The formed categories were carefully revised to come to the final ones that would facilitate the aim of the research without being complex. In a following step, interpretive data acquired from the interviews in general were considered through the authors’ critical perspective on the excerpts [71] and were added in the thematic categories which were classified in relation to the interviews questions as a wider framework of the research questions to reach easier the study’s aim and the interpretation of the emerged issues initiated. As this procedure was completed, the authors concluded to the six (6) themes that were regarded as the essence of the research, naming them accordingly while excerpts from the data were transferred to them to offer a clear understanding for each one of them [74]. The final stage was to form the academic report which involved discussion and interpretation of the data to explain the aspects of the study. The added interview excerpts were translated from Greek to English and the authors tried not to lose distinctions of the context.
Even with some changes and improvements from the previous version, I think that this paper still needs to be improved at the scientific aspect, in order to be published.

Round 3
Reviewer 3 Report
Dear colleagues
In general, the paper is interesting, and it improved from the initial version. However, in my opinion, there are some fundamental scientific flaws:
- By doing 23 interviews it would important to use qualitative analysis software to get stronger results.
- The research questions are still not directly supported by the state of art. What are the authors that lead to state RQ1? 2, 3 and 4?
When was the seminar conducted and the interviews performed? What was the time lag?
There are some issues that still need some scientific validation. If those issues were addressed I think that the paper could be published. Meanwhile, I will keep my initial decision.
Anyway, I think that the authors can take an effort to reorganize the paper including these ideas and then might try to publish the paper. I think that this is very close to be published.